

# Comparative Analysis of Compact Portable and Indoor Rainfall Simulators

Raquel N. R. Falcão[1], Josef Krása[1], Martin Neumann[1], Jan-František Kubát[1], Corinna Gall[2], Steffen Seitz[2]

[1] CTU in Prague, Department of Landscape Water Conservation, Faculty of Civil Engineering, Thakurova 7, 166 29 Prague 6, Czech Republic

[2] Soil Science and Geomorphology, Department of Geoscience, University of Tübingen, Rümelinstrasse 19–23, 72070 Tübingen, Germany

*Correspondence to*: Raquel N. R. Falcão (raquel.nogueira.rizzotto.falcao@fsv.cvut.cz)

**Abstract**

Rainfall simulators have been widely used and are indispensable in soil hydrology and erosion research. Although rainfall simulators can be used to address various research questions, there is no standardized methodology; they differ in design and rainfall characteristics. The present study aims to describe the design and testing of five rainfall simulator setups that vary notably in weight, volume, and transportability. Additionally, this article seeks to clarify procedural aspects involved in conducting a rainfall simulation in the field. The following parameters are used to compare the simulators: drop size distribution and terminal velocity, uniformity of the spatial distribution of raindrops over the sprinkled surface area (Christiansen uniformity coefficient; CU), kinetic energy (KE), and rainfall intensity. The Thies laser disdrometer and Tübingen Splash Cups (T-cups) were used to measure raindrop's KE to identify similarities and differences in their rainfall characteristics.

The rainfall simulator setups produce rainfall intensities ranging from 28 to 95 mm h$^{-1}$, with CU values ranging from 60.5% to 75.8%. More than 90% of measured drops were slower than 3.8 m s$^{-1}$ for all simulations. The maximum number of drops was below 0.5 mm class, generally smaller than that observed in natural rain, and all at 1.4-1.8 m s$^{-1}$ velocity. We found that kinetic energy (KE) measured with T-cups agreed with values calculated with the Thies disdrometer, confirming its relevance in rainfall studies. Indoor simulator setups produced the highest KEs, whereas the portable systems showed considerably lower values.

This study emphasizes the importance of accurately characterizing rainfall parameters before soil erosion measurements. Rain simulators are then a powerful tool in erosion research. The presented methodologies and insights provide means for improved assessment of soil erosion risks, particularly regarding their practicality in remote areas.

**Keywords:** Rainfall simulators, Soil erosion, Drop size distribution, Drop velocity, Raindrop kinetic energy



**Graphical abstract**

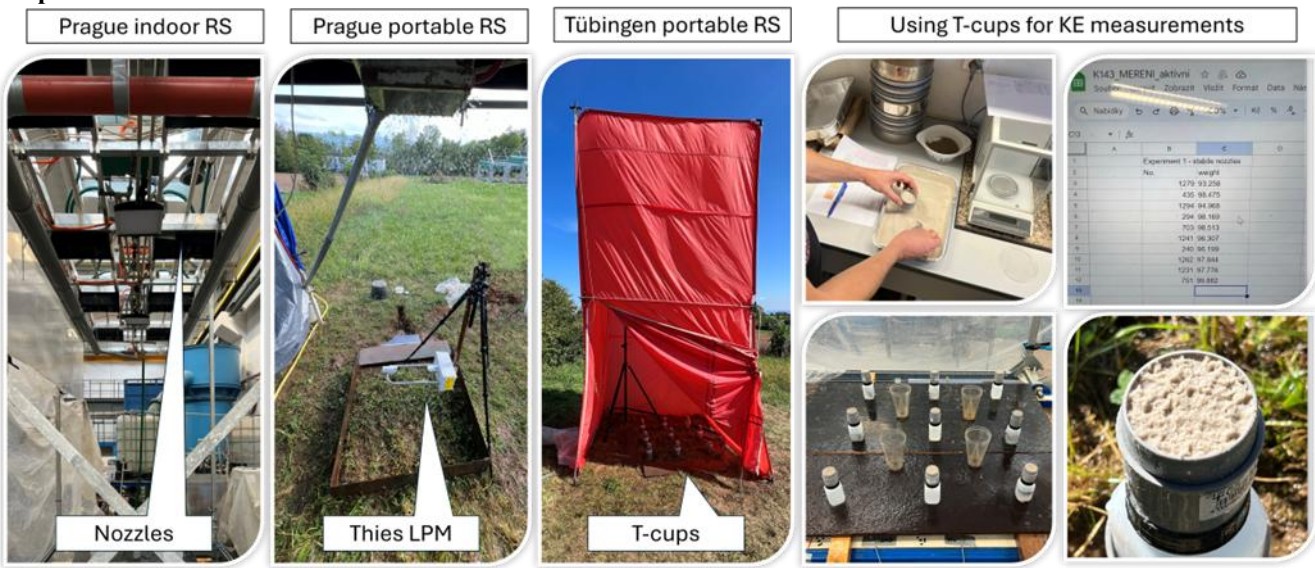



## 1    Introduction

Since its first use at the end of the 19th century by Ewald Wollny (Dotterweich, 2013), rainfall simulators have been used
extensively for various research questions in geomorphology, soil science, hydrology, and related research areas (Cerdá, 2013;
Luz et al., 2024; Mutchler & Hermsmeier, 1965). Over the years, they have become an indispensable tool worldwide, mainly
for studying soil hydrology and erosion processes, as they allow a precise and reproducible assessment of the different factors
influencing these phenomena (Cerdá, 2013; Iserloh et al., 2013). Rainfall simulators offer a substantial advantage by enabling
the maintenance of specific rainfall characteristics while allowing adjustments based on the research question. On the one
hand, this makes it possible to investigate how alterations in rainfall characteristics affect soil erosion processes (e.g., rainfall
intensity (Meyer, 1981), duration (Zhao et al., 2021), raindrop kinetic energy (Fernández-Raga et al., 2010; Goebes et al.,
2014), rain temperature (Sachs & Sarah, 2017), while on the other hand, the influence of various soil properties on soil erosion
is analyzed under constant rainfall characteristics (e.g., parent material (Rodrigo-Comino et al., 2018), soil aggregate stability
(Le Bissonnais, 1996), or antecedent soil moisture (Le Bissonnais et al., 1995).

Due to this wide range of possible applications, rainfall simulators are not standardized. Rainfall simulations usually differ
clearly in plot size and shape, rainfall intensity, spatial rainfall distribution, and raindrop characteristics, among others (Iserloh
et al., 2013). However, we can distinguish between transportable (Clarke & Walsh, 2007; Iserloh et al., 2012) and stationary
rainfall simulators (Lassu et al., 2015; Nanko et al., 2008), which can be further differentiated according to their method of
forming raindrops: gravitational rainfall simulators, which generate raindrops solely by the hydrostatic pressure created by a
column of water above the outlets, and pressurized rainfall simulators, which form raindrops using water pressure and different
types of nozzles (Kavka & Neumann, 2021; Koch et al., 2024). Although transportability is often a crucial factor for fieldwork,
it can substantially constrain the potential of droplet formation compared to controlled laboratory conditions.

The primary goal of rainfall simulators is to create consistent rainfall conditions that allow for scientific repetition of
measurements. While these simulators strive to approximate the characteristics of natural rainfall, they face technical
limitations, such as replicating the exact spectrum of raindrop size distribution or terminal velocities (Assouline et al., 1997;
Iserloh et al., 2012; Villermaux & Bossa, 2009). Therefore, while the aim is to be as "near-natural" as possible, the focus
remains on achieving steady, controllable rainfall parameters for research. A particular difficulty arises from natural rainfall
being subject to fluctuations in rainfall intensity, which is not reflected in rainfall simulations that generally employ constant-
intensity rainfall (Dunkerley, 2008, 2021). This is a disadvantage of rainfall simulations and a significant advantage for
answering experimental questions more reliably without fluctuating rainfall conditions. Regarding the generation of natural
rainfall conditions, the following vital parameters have emerged over time as the most suitable for comparing different rainfall
simulators: drop size distribution and terminal velocity, uniformity of the spatial distribution of raindrops over the sprinkled
surface area, the overall rainfall intensity, and kinetic energy (Fister et al., 2012; Kavka & Neumann, 2021; Koch et al., 2024;
Ries et al., 2009; Villermaux & Bossa, 2009).





One of the most common instruments for measuring raindrop characteristics is laser disdrometers, such as the Laser Precipitation Monitor (LPM) by Thies, which can record the size distribution and fall velocity of raindrops as well as rainfall intensity and amount (Lanzinger et al., 2006; Ries et al., 2009). Click or tap here to enter text.Another commonly used factor is the Christiansen uniformity coefficient (CU; Christiansen, 1942), for which rainfall is collected with rain collectors in a uniform grid on the sprinkled surface area  for a certain period (Keller & Bliesner, 1990; Little et al., 1993). Click or tap here

to enter text.

    Furthermore, the raindrops' kinetic energy disrupts soil aggregates and detaches soil particles at the soil surface, making it a key parameter in soil erosion research. Nonetheless, it is not routinely measured (Petrů & Kalibová, 2018). The kinetic energy of raindrops can either be calculated from rainfall intensities as described in Iserloh et al. (2013) or determined using splash cups, which build on the principle of measuring sand loss caused by droplet impact, e.g., Tübingen Splash Cups (T-cups,

Scholten et al., 2011). In a comparative study of splash erosion devices, the T-cups proved to be the best device for determining the kinetic energy of rainfall (Fernández-Raga et al., 2019) and highly suitable to supplement comparative rainfall studies. Splash cups have been used for a variety of purposes, such as monitoring splash erosion under natural rainfall (Laburda et al., 2021), the effect of vegetation cover on soil erosion and kinetic energy (Goebes et al., 2015; Mosley, 1982; Shinohara et al., 2018; Vis, 1986), and wind erosion studies (Cornelis et al., 2004; Erpul et al., 2005).

This study describes the design of three rainfall simulators: the Prague indoor rainfall simulator, the Prague portable rainfall simulator, and the Tübingen portable rainfall simulator, which differ in terms of transportability, nozzle type, and type of raindrop generation. The first two rainfall simulators were each installed with two different nozzles, resulting in a total of five different rainfall simulator setups. We used the same test setup consisting of the Thies LPM and rain collectors for all five rainfall simulator setups to compare their artificial rainfall using the standard parameters described above. Additionally, we

used T-cups to quantify the kinetic energy of the artificial rainfall, whereby this method was applied here for the first time to compare different rainfall simulators with each other. This study's primary objective was to identify the similarities or differences between the three rainfall simulators, respectively the five rain simulator setups, regarding the rainfall characteristics investigated. Additionally, this article seeks to clarify procedural aspects involved in conducting a rainfall simulation in the field. By offering insights into tasks and requirements, we hope to assist researchers considering conducting

this type of experiment for the first time. The information is intended as a practical guide, facilitating a smoother and more informed approach to implementing rainfall simulations.





## 2    Equipment Design and Installation

### 2.1    Rainfall simulators

We used three rainfall simulators for this study, two of them provided by CTU Prague, Czech Republic, and one by the University of Tübingen, Germany (Table 1). The first two were installed using two different nozzles, totalling five different setups:

**Table 1: Main characteristics of rainfall simulators**

| Setup n° | Device | Plot design | Drop Falling height (m) | Nozzle |
|---|---|---|---|---|
| 1 | Prague Indoor RS | 4 × 1 m, rectangular | 2.6 | Veejet 80100 (swing system) |
| 2 | Prague Indoor RS | 4 × 1 m, rectangular | 2.6 | WSQ40 (pulse system) |
| 3 | Prague Portable RS | 1 × 1 to 2 × 2 m rectangular | 2 | Veejet 9550 (swing system) |
| 4 | Prague Portable RS | 1 × 1 to 2 × 2 m rectangular | 2 | WSQ40 (pulse system) |
| 5 | Tübingen Portable RS | 0.4 × 0.4 to 1 × 1 m, rectangular | 3.5 | Lechler 490.808.30.CE (continuous pressure system) |

### 2.1.1    Prague Indoor Rainfall Simulator (pulse and swing system of the nozzle)

The Prague Indoor Rainfall Simulator was built at the Czech Technical University (CTU) in 1999, and has undergone various improvements throughout its lifespan (Figure 1). Now, the RS provides conditions for various types of experiments on an experimental plot with a size of 4 × 1 m, as described in Kavka et al. (2019). The soil container can be elevated to create slopes up to 35°. In addition, the simulator is equipped with a freezing device, enabling the study of the effects of freezing and thawing cycles over a temperature range of -15°C to 40°C. The solid bottom of the experimental plot is divided into four segments,
each perforated by holes to collect percolated water and monitor water infiltration into the deeper soil layers.



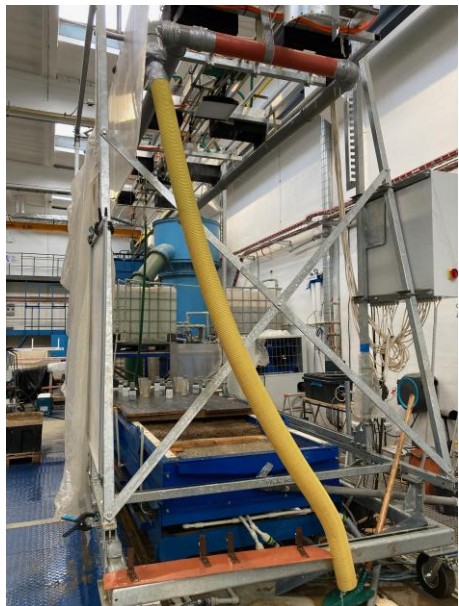

**Figure 1: Prague Indoor Rainfall Simulator**

There are two different methods of producing rainfall over the plot. Both consist of nozzles with water under pressure. The first method is a swing system where nozzles with swing movement spray water on the surface. The amount and speed of the

swings set the rainfall intensity. The second pulse system used square nozzles, which are stationary, and the rainfall intensity was set by the interruption of rainfall. The stationary system was also used for the small rainfall simulator of CTU in Prague. Rainfall intensity could be set at 20-80 mm h$^{-1}$ for the swing system and 20-160 mm h$^{-1}$ for the pulse system.

The container with a soil sample is placed beneath the RS. The setup consists of a 5 cm sand layer at the bottom topped by a 15 cm layer of soil. One of the key advantages of this device is its accessibility to water, electricity, and other necessary

facilities. The experiment can be conducted during rainfall or winter when field experiments are not feasible. However, a drawback is that the soil sample analysed is always disturbed, and experiments involving crops are not possible.

### 2.1.2 Prague Portable Rainfall Simulator (pulse and swing system of the nozzle)

The Czech Technical University (CTU) in Prague built the portable rainfall simulator in 2020. After the experience of conducting experiments with large field and indoor rainfall simulators, there was a lack of a smaller device that allows access

to localities that are not possible to reach with a large rainfall simulator (e.g., forest, mountains) and are less dependent on human resources and amount of water supply (Figure 2). Since 2021, the small RS has been used for routine experiments on various types of soil covers. A detailed description of the calibration is provided in Kavka & Neumann (2021).

The simulator is supported by a durable 3 x 3 m commercial tent. A step engine and computer program control a single movable, replaceable nozzle, while a solenoid valve manages water inflow. The system is equipped with an electric water

pump, a manual pressure valve to adjust the required pressure, a water filter, and a bypass hose to ensure smooth operation.



Additionally, the rain control unit enables the configuration of experimental parameters, offering options for pulse or swing systems, adjusted rainfall intensity, and nonlinear rainfall settings.

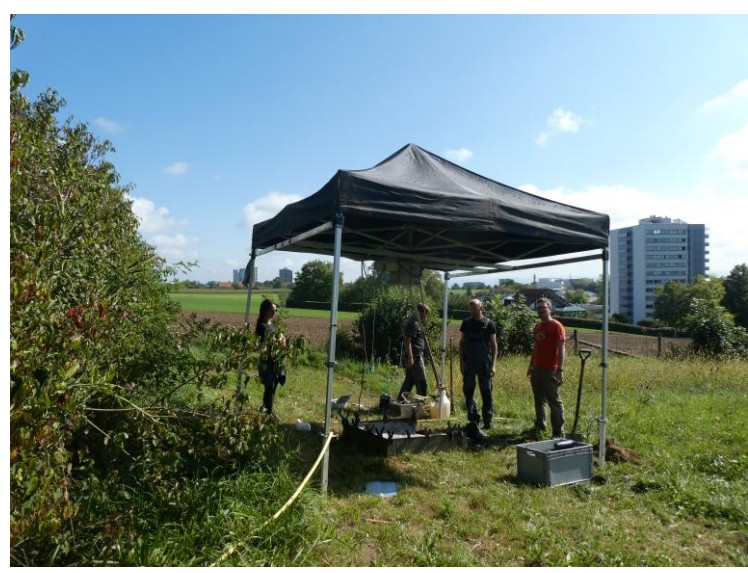

Figure 2: Prague Portable Rainfall Simulator

This rainfall simulator can use swing or pulse systems as needed. This rainfall simulator uses both systems. During this experiment, a swing system was applied for both nozzles tested. To produce rainfall, one replaceable nozzle is used, which is placed above the center of the plot with size 1 × 1 m (up to the maximum size of 2 × 2 m), at two meters height. Kinetic energy could be changed by using different nozzles, but the individual calibration to achieve the spatial distribution has been done for every nozzle. The amount and duration of pauses in rainfall by closing the solenoid valve on the water inlet is used to set

rainfall intensity. The rainfall intensity was set to 1 liter per square meter, corresponding to 60 mm h⁻¹, with water consumption ranging from approximately 150 to 200 liters per hour. The higher water consumption can be attributed to the nozzle spray pattern and an overspray around the plot.

Wind can adversely affect the spatial distribution of rainfall, so we shielded the experimental plot with a plastic tarpaulin.

### 2.1.3 Tübingen Portable Rainfall Simulator (continuous pressure system)

The Tübingen Portable Rainfall Simulator (TRS) was developed by the Chair of Soil Science and Geomorphology at the University of Tübingen in 2010 and first described in Iserloh et al. (2013). The design of the Tübingen Portable RS focused, in particular, on portability in the field so that it can be used worldwide with minimal staff and in difficult-to-reach terrain (Figure 3). Thus, the Tübingen Portable RS has proven its reliability in various ecosystems globally such as temperate and subtropical forests, semiarid and arid deserts or different farming trials on three continents (Riveras-Muñoz et al., 2015). It

features a lightweight, highly rugged shelter made from tear-resistant tent fabric (Hilleberg AB, Norway), with an internal volume of 16 m³, a 2 × 2 m ground area, and a height of 4 m, supported by aluminum poles. The system includes a single



nozzle system setup, using a standard Lechler 490.808.30.CE nozzle (Lechler GmbH, Germany) mounted on a lightweight aluminium tripod of adjustable height, allowing the nozzle to be positioned at 3.5 m. Additionally, the simulator is equipped with a portable electric pressure water pump (FloJet Quiet Quad, Xylem, USA) powered by a 12 V LiFePo battery, and a
portable plastic tank, ensuring pressure of 60 kPa at the nozzle.

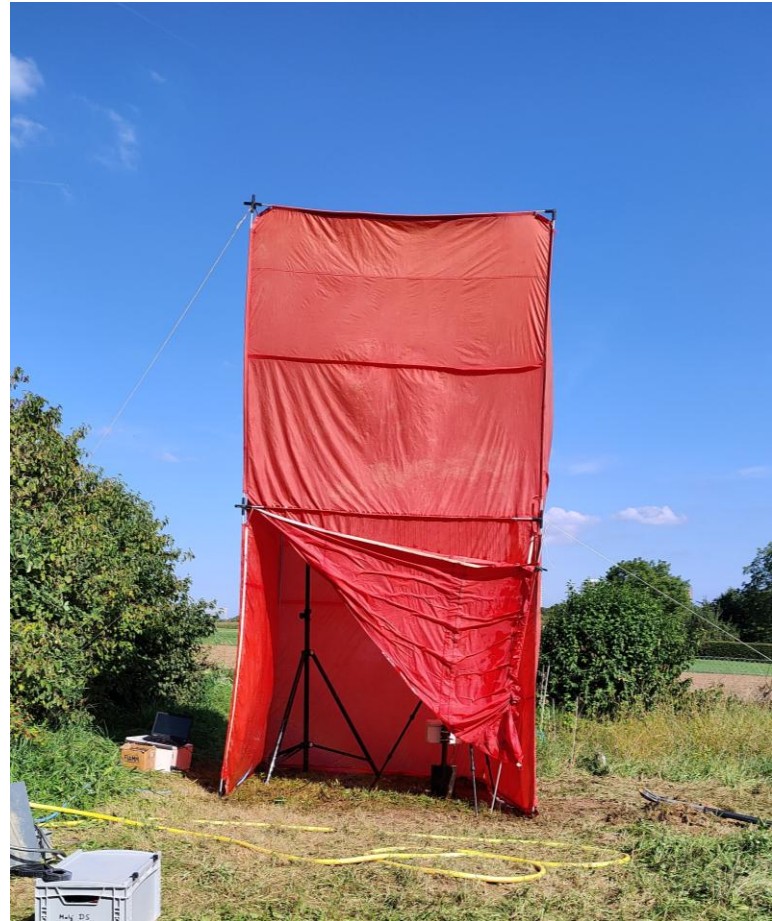

**Figure 3: Tübingen Portable Rainfall Simulator**

Compared to the Prague Portable RS, the nozzle of the Tübingen Portable RS produces raindrops continuously without a pulse or swing system and generally has a simpler structure. The rainfall intensity can be controlled by changing the nozzle's pressure
or, if necessary, by varying the nozzles. However, to guarantee comparability of measurements, a standard Lechler nozzle is commonly used, which achieves a rainfall intensity of 45 mm h$^{-1}$. When operating the Tübingen Portable RS in the field, it is leveled out depending on the terrain and slope and can be used with $1 \times 1$ m or micro-scale $0.4 \times 0.4$ m runoff plots. The runoff plots are made of stainless steel, with the frames connected on three sides. When installed in the topsoil, they are combined with a triangular surface runoff gutter and covered with a metal sheet during rainfall simulations.





## 2.2    Assessing rainfall attributes

### 2.2.1    Disdrometer

The Laser Precipitation Monitor 5.4110 (LPM) by Thies Clima (Göttingen, Germany) is a state-of-the-art optical disdrometer measuring the drop size distribution (DSD) with number, diameter, and velocities of various droplets (Lanzinger et al., 2006). It registers individual drops with diameters varying from 0.125 to 8 mm in 22 categories and velocities from 0 to 10 m s$^{-1}$ in 20 categories.

LPM records drop size and drop velocity classes, so kinetic energy expenditure (KE$_r$, J m$^{-2}$ h$^{-1}$), kinetic energy (KE, J m$^{-2}$ mm$^{-1}$), and median volumetric drop diameter (d$_{50}$) were computed by utilizing the average value within each class, similar to Iserloh et al. (2013). For the calculations, the drops are assumed to be spherical, and for every 1-min period, KE$_R$ can be calculated as:

$$KE_R = \left(\frac{\pi}{12}\right)\left(\frac{1}{10^6}\right)\left(\frac{3600}{t}\right)\left(\frac{1}{A}\right)\sum_{i=1}^{22} n_i D_i^3 (v_{D_i})^2 \qquad \text{1}$$

$$KE = \left(\frac{KE_R}{i}\right) \qquad \text{2}$$

Where A is the sampling area of the LPM (44,1 cm$^2$), $n_i$ is the number of drops of diameter Di, $v_{Di}$ is the measured fall velocity of the drop with diameter Di, and t = 60 s, adapted from Fornis et al., 2005.

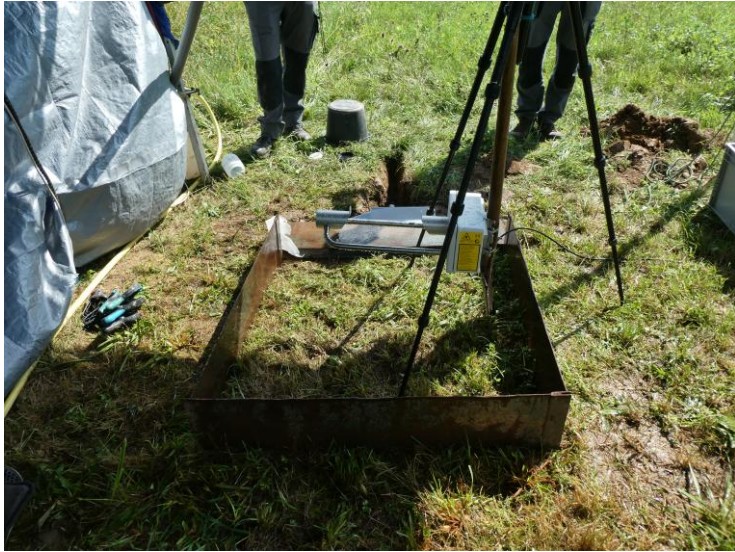

**Figure 4: Thies disdrometer measurement with the Prague Portable Rainfall Simulator**



### 2.2.2    Splash cups

The kinetic energy of rainfall was additionally measured using Tübingen Splash Cups (T-cups; Scholten et al., 2011). The measuring surface is punctiform with a diameter of 46 mm (16.62 cm$^2$ of surface area) 2.6 times smaller than the laser beam of the LPM (Figure 5). Splash cups were filled with sieved uniform sand (0.125 - 0.200 mm), and the detached material was calculated by the difference of dry sand weights before and after each rainfall simulation. The total detached sand (ds) per cup was converted to the kinetic energy of rainfall using a linear function provided by Scholten et al. (2011):

$$KE_{rf}\left(J/_{m^2}\right) = ds_{sc}(g) \times 0.1455 \times \left(\frac{10\,000\,(cm^2)}{\pi\,r^2{}_{sc}}\right) \qquad 3$$

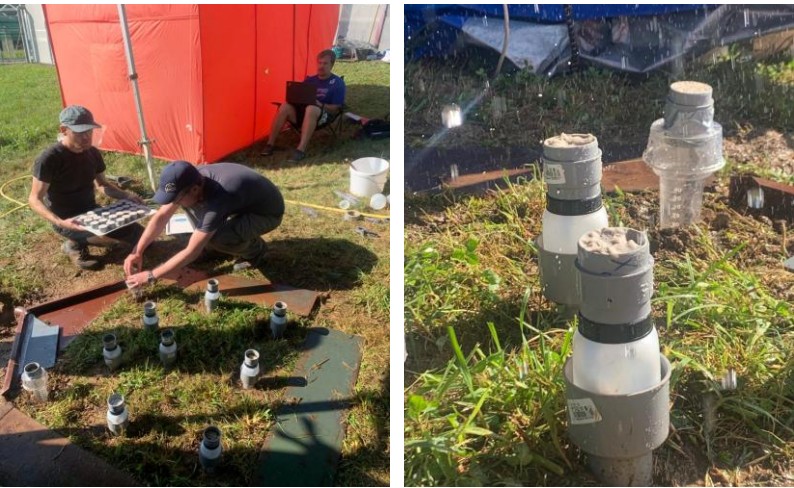

**Figure 5: Splash cups installation prior to the experiment (left) and during the simulation (right)**

The resulting KE for each experiment was averaged over the plot by Inverse Distance Weighing (IDW) interpolation.

### 2.3    Spatial rainfall distribution and measurement design

The homogeneity of the rainfall was analyzed by changing the positioning of the LPM over the plot area, or by covering the test plot with splash cups (Figure 6). The center of the measured area was placed above the defined positions.





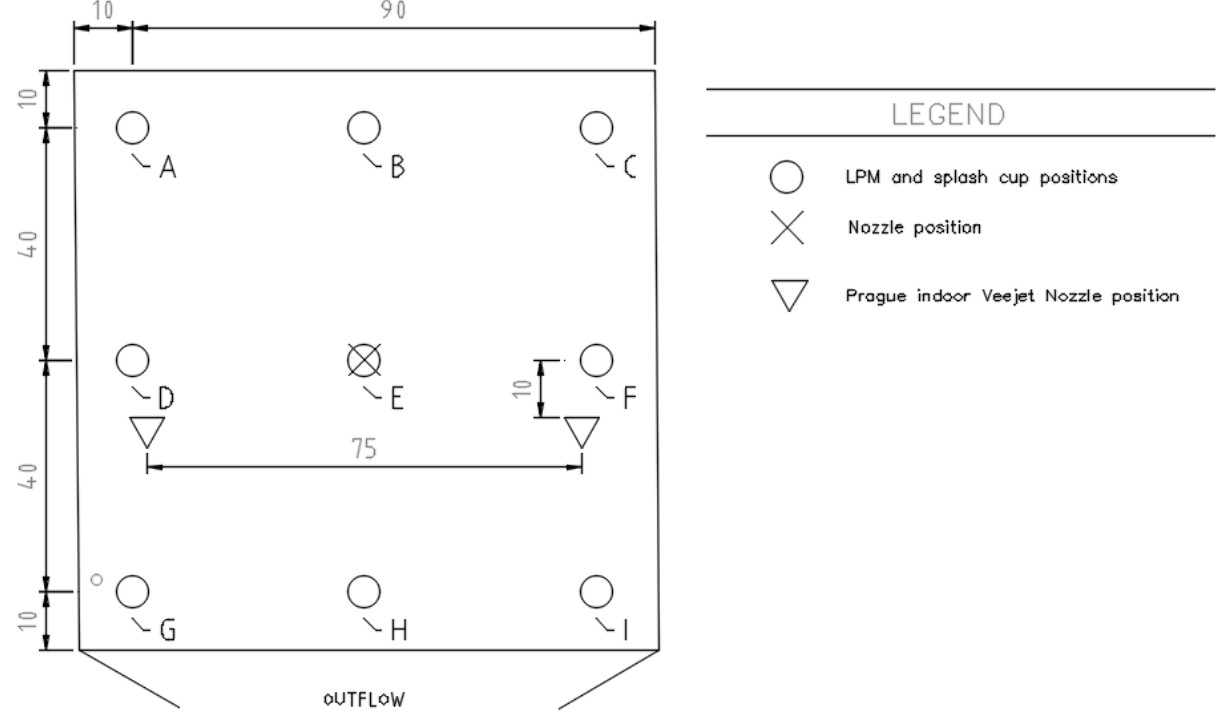

**Figure 6: top view of the test setup**

For the portable RS, the plot area was divided into a regular grid, and nine positions were used to analyze drop spectra with
the LPM (Figure 6). Five one-minute-long interval replicates were performed at each position, and the mean values of these
five values were used. The same plot was used for the CTU laboratory RS.

### 2.4    Statistical approach and data evaluation

This study's data handling and graphical representations were performed using PyCharm IDE in Python 3.9.7 and R 4.2.2 (R
Core Team, 2022).

The standard deviation of rain intensity heatmaps was generated using IDW interpolation by averaging the values for each
position, applied to a 15 × 15 point grid within the 1 × 1 m measured area.

D50 was calculated both as a count and a mass, using the midpoint of the size bins.

To assess whether the distributions of generated drops and KEr values differed across equipment, we used the Kruskal-Wallis
test with a 95% confidence level. Where significant effects were detected, pairwise comparisons using post-hov Dunn's test
was done. Analyses were conducted with IBM SPSS, version 29.0.0.0.





## 3    Results

The rainfall simulators produced varying intensities from 28 to 95 mm h-1 (Table 2), as measured by the Thies LPM. The Tübingen Portable RS has the lowest standard deviation of rainfall intensity (

Table A1). The results further showed very different, and in some cases high, intensities for the Prague simulator setups, with

the indoor simulators reaching the highest kinetic energies above 700 J m-2 h-1. Thies disdrometers have consistently underestimated the KE with high drop numbers.

**Table 2: Main results of rainfall characteristics for each RS: mean intensity [I], Christiansen Uniformity [CU], spatial coefficient of variation [CV] estimated as the ratio of the standard deviation to the mean intensity, mean drop number [n] per minute, median**
**volumetric drop diameter [d50] per count and mass, mean kinetic energy expenditure [KE$_R$], mean kinetic energy per unit area per unit depth of rainfall [KE], as measured by LPM.**

| n° | Device | I (mm h$^{-1}$) | CU (%) | CV (%) | n (min$^{-1}$) | d50 count (mm) | d50 mass (mm) | KE$_R$ (J m$^{-2}$ h$^{-1}$) | KE (J m$^{-2}$ mm$^{-1}$) |
|---|---|---|---|---|---|---|---|---|---|
| 1 | Prague indoor RS Veejet 80100 | 65 ± 20 | 60.5 | 29.0 | 17 565 | 0.29 | 2.37 | 721 ± 255 | 11.03 ± 2.04 |
| 2 | Prague indoor RS WSQ40 | 95 ± 28 | 75.8 | 28.8 | 15 762 | 0.40 | 2.16 | 753 ± 260 | 7.79 ± 0.72 |
| 3 | Prague portable RS Veejet 9550 | 64 ± 22 | 68.1 | 33.0 | 10 668 | 0.42 | 1.58 | 285 ± 91 | 4.49 ± 0.62 |
| 4 | Prague portable RS WSQ40 | 65 ± 27 | 62.9 | 40.0 | 10 249 | 0.37 | 2.41 | 416 ± 257 | 6.05 ± 1.35 |
| 5 | Tübingen portable RS | 28 ± 5 | 70.9 | 11.5 | 19 898 | 0.36 | 0.67 | 77 ± 17 | 2.78 ± 0.67 |

We used the Kruskal-Wallis nonparametric test for the total number of generated drops, as the data did not follow a normal distribution (Shapiro-Wilk, $p<0.05$). We found $H(4) = 115.59$, $p < 0.05$, indicating significant differences among the groups, with 4 degrees of freedom. Results of the pairwise comparison are in Table A2.

The number of drops generated by each device varies strongly, with the Tübingen Portable RS having more than double the count of drops than the Prague portable system (Figure 7). For the Prague indoor system, the number of drops varies substantially with the installed nozzle system, with the swiping system (Veejet) showing the highest variation over all devices.



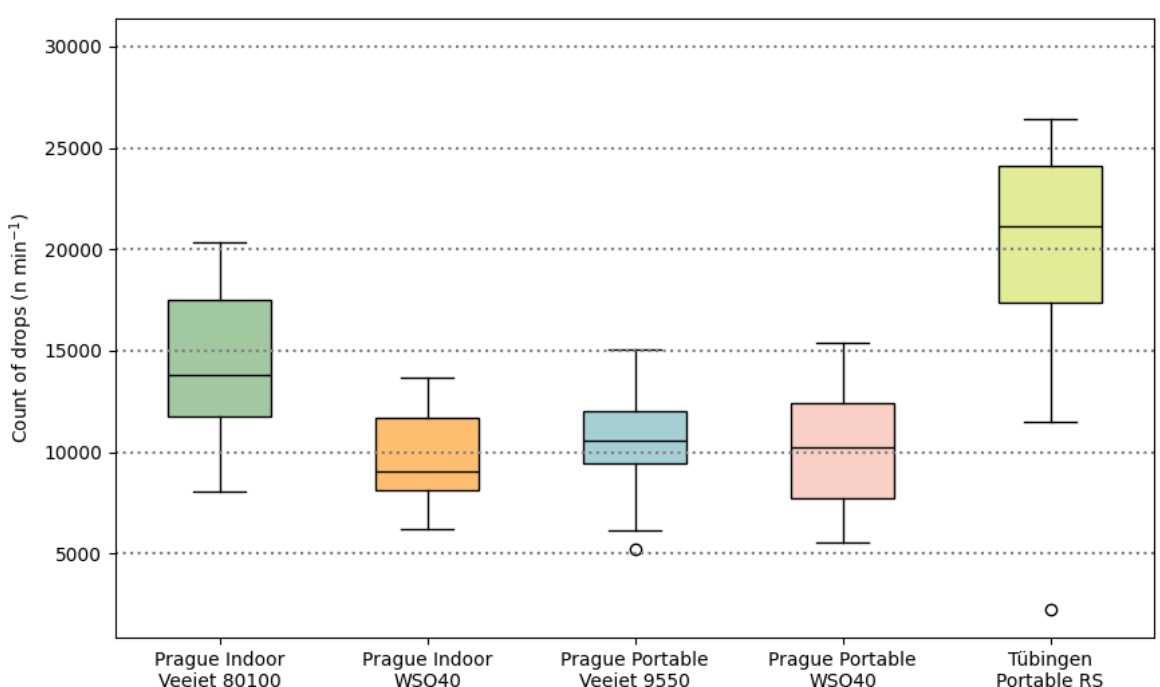

**Figure 7: Distribution of the number of drops generated by the five rainfall simulators used in the Kruskal-Wallis test (n=25)**

### 3.1 Drop spectra

The drop size and fall velocity measured with the LPM show distinct differences across devices (Figure 8). The maximum drop size was below 0.5 mm for Equipment 1, 4, and 5, and between 0.5 and 0.75 mm for Equipment 2 and 3. The maximum drop number had a fall velocity between 1.0 – 1.4 m s-1 for Equipment 1, 2, and 3, and at 1.4-1.8 m s$^{-1}$ velocity for Equipment 4 and 5. The raindrop size distribution shows significant skewness toward lower fall speeds and smaller diameters: in all simulations, over 80% of the total drops are < 1 mm, and over 90% in indoor simulations. The skewness of the drop size distribution is shown in the appendix (Figure A1).





**Equipment 1: Prague Indoor RS Veejet 80100**  total drop number **14475**

| drop fall velocity (m.s-1) | <0.5 | 0.5-1.0 | 1.0-1.5 | 1.5-2.0 | 2.0-2.5 | 2.5-3.0 | 3.0-3.5 | 3.5-4.0 | 4.0-4.5 | 4.5-5.0 | 5.0-5.5 | 5.5-6.0 | >6.0 |
|---|---|---|---|---|---|---|---|---|---|---|---|---|---|
| >10.0 | 0 | 0 | 0 | 0 | 0 | 0 | 0 | 0 | 0 | 0 | 0 | 0 | 0 |
| 9.0-10.0 | 1 | 0 | 0 | 0 | 0 | 0 | 0 | 1 | 0 | 0 | 0 | 0 | 0 |
| 8.2-9.0 | 1 | 0 | 0 | 0 | 0 | 1 | 1 | 1 | 0 | 0 | 0 | 0 | 0 |
| 7.4-8.2 | 2 | 0 | 0 | 0 | 3 | 3 | 2 | 1 | 1 | 0 | 0 | 0 | 0 |
| 6.6-7.4 | 4 | 0 | 0 | 7 | 11 | 6 | 3 | 1 | 1 | 0 | 0 | 0 | 0 |
| 5.8-6.6 | 8 | 1 | 9 | 27 | 17 | 6 | 2 | 2 | 1 | 0 | 0 | 0 | 0 |
| 5.0-5.8 | 17 | 7 | 61 | 49 | 12 | 7 | 4 | 2 | 1 | 1 | 0 | 0 | 0 |
| 4.2-5.0 | 51 | 73 | 139 | 34 | 16 | 9 | 3 | 1 | 1 | 0 | 0 | 0 | 1 |
| 3.4-4.2 | 145 | 268 | 113 | 26 | 15 | 6 | 2 | 1 | 1 | 0 | 0 | 0 | 1 |
| 3.0-3.4 | 155 | 216 | 27 | 13 | 6 | 2 | 1 | 0 | 0 | 0 | 0 | 0 | 0 |
| 2.6-3.0 | 301 | 313 | 21 | 14 | 5 | 2 | 1 | 1 | 0 | 0 | 0 | 0 | 0 |
| 2.2-2.6 | 660 | 391 | 31 | 12 | 5 | 2 | 1 | 0 | 0 | 0 | 0 | 0 | 0 |
| 1.8-2.2 | 1480 | 318 | 35 | 12 | 5 | 1 | 1 | 0 | 0 | 0 | 0 | 0 | 0 |
| 1.4-1.8 | 2117 | 317 | 40 | 12 | 3 | 1 | 1 | 0 | 0 | 0 | 0 | 0 | 0 |
| 1.0-1.4 | 2427 | 445 | 40 | 10 | 3 | 1 | 0 | 0 | 0 | 0 | 0 | 0 | 0 |
| 0.8-1.0 | 1387 | 210 | 18 | 4 | 1 | 0 | 0 | 0 | 0 | 0 | 0 | 0 | 0 |
| 0.6-0.8 | 1076 | 181 | 14 | 3 | 1 | 0 | 0 | 0 | 0 | 0 | 0 | 0 | 0 |
| 0.4-0.6 | 577 | 126 | 10 | 2 | 0 | 0 | 0 | 0 | 0 | 0 | 0 | 0 | 0 |
| 0.2-0.4 | 151 | 49 | 4 | 1 | 0 | 0 | 0 | 0 | 0 | 0 | 0 | 0 | 0 |
| 0-0.2 | 10 | 3 | 1 | 0 | 0 | 0 | 0 | 0 | 0 | 0 | 0 | 0 | 0 |

Drop diameter (mm)

**Equipment 2: Prague Indoor RS WSQ40**  total drop number **9606**

| drop fall velocity (m.s-1) | <0.5 | 0.5-1.0 | 1.0-1.5 | 1.5-2.0 | 2.0-2.5 | 2.5-3.0 | 3.0-3.5 | 3.5-4.0 | 4.0-4.5 | 4.5-5.0 | 5.0-5.5 | 5.5-6.0 | >6.0 |
|---|---|---|---|---|---|---|---|---|---|---|---|---|---|
| >10.0 | 0 | 0 | 0 | 0 | 0 | 0 | 0 | 0 | 0 | 0 | 0 | 0 | 0 |
| 9.0-10.0 | 0 | 0 | 0 | 0 | 0 | 0 | 0 | 0 | 0 | 0 | 0 | 0 | 0 |
| 8.2-9.0 | 0 | 0 | 0 | 0 | 0 | 0 | 0 | 0 | 0 | 0 | 0 | 0 | 0 |
| 7.4-8.2 | 0 | 0 | 0 | 0 | 0 | 0 | 0 | 0 | 0 | 0 | 0 | 0 | 0 |
| 6.6-7.4 | 0 | 0 | 0 | 0 | 0 | 0 | 0 | 0 | 0 | 0 | 0 | 0 | 0 |
| 5.8-6.6 | 2 | 0 | 0 | 1 | 5 | 3 | 1 | 1 | 1 | 1 | 0 | 0 | 0 |
| 5.0-5.8 | 5 | 0 | 8 | 29 | 11 | 7 | 4 | 2 | 2 | 1 | 1 | 1 | 0 |
| 4.2-5.0 | 17 | 16 | 121 | 57 | 27 | 14 | 7 | 4 | 3 | 2 | 1 | 0 | 0 |
| 3.4-4.2 | 76 | 241 | 249 | 78 | 35 | 16 | 8 | 4 | 3 | 1 | 1 | 0 | 0 |
| 3.0-3.4 | 75 | 300 | 87 | 35 | 15 | 7 | 3 | 1 | 1 | 0 | 0 | 0 | 0 |
| 2.6-3.0 | 136 | 415 | 46 | 23 | 12 | 6 | 2 | 1 | 1 | 0 | 0 | 0 | 0 |
| 2.2-2.6 | 327 | 501 | 45 | 24 | 10 | 5 | 2 | 1 | 0 | 0 | 0 | 0 | 0 |
| 1.8-2.2 | 773 | 311 | 63 | 25 | 10 | 4 | 1 | 1 | 0 | 0 | 0 | 0 | 0 |
| 1.4-1.8 | 1067 | 280 | 67 | 23 | 7 | 3 | 1 | 0 | 0 | 0 | 0 | 0 | 0 |
| 1.0-1.4 | 1417 | 401 | 61 | 17 | 5 | 2 | 1 | 0 | 0 | 0 | 0 | 0 | 0 |
| 0.8-1.0 | 633 | 169 | 24 | 6 | 2 | 1 | 0 | 0 | 0 | 0 | 0 | 0 | 0 |
| 0.6-0.8 | 474 | 132 | 17 | 4 | 1 | 0 | 0 | 0 | 0 | 0 | 0 | 0 | 0 |
| 0.4-0.6 | 249 | 81 | 12 | 3 | 1 | 0 | 0 | 0 | 0 | 0 | 0 | 0 | 0 |
| 0.2-0.4 | 65 | 33 | 5 | 1 | 0 | 0 | 0 | 0 | 0 | 0 | 0 | 0 | 0 |
| 0-0.2 | 4 | 1 | 0 | 0 | 0 | 0 | 0 | 0 | 0 | 0 | 0 | 0 | 0 |

Drop diameter (mm)





**Equipment 3: Prague Portable RS Veejet 9550**    total drop number    **10745**

| drop fall velocity (m s-1) | <0.5 | 0.5-1.0 | 1.0-1.5 | 1.5-2.0 | 2.0-2.5 | 2.5-3.0 | 3.0-3.5 | 3.5-4.0 | 4.0-4.5 | 4.5-5.0 | 5.0-5.5 | 5.5-6.0 | >6.0 |
|---|---|---|---|---|---|---|---|---|---|---|---|---|---|
| >10.0 | 0 | 0 | 0 | 0 | 0 | 0 | 0 | 0 | 0 | 0 | 0 | 0 | 0 |
| 9.0-10.0 | 0 | 0 | 0 | 0 | 0 | 0 | 0 | 0 | 0 | 0 | 0 | 0 | 0 |
| 8.2-9.0 | 1 | 0 | 0 | 0 | 1 | 0 | 0 | 0 | 0 | 0 | 0 | 0 | 0 |
| 7.4-8.2 | 3 | 0 | 1 | 2 | 2 | 1 | 0 | 0 | 0 | 0 | 0 | 0 | 0 |
| 6.6-7.4 | 8 | 2 | 7 | 8 | 3 | 1 | 0 | 0 | 0 | 0 | 0 | 0 | 0 |
| 5.8-6.6 | 22 | 14 | 30 | 15 | 4 | 1 | 0 | 0 | 0 | 0 | 0 | 0 | 0 |
| 5.0-5.8 | 59 | 71 | 68 | 19 | 4 | 1 | 1 | 0 | 0 | 0 | 0 | 0 | 0 |
| 4.2-5.0 | 140 | 207 | 98 | 19 | 6 | 2 | 1 | 0 | 0 | 0 | 0 | 0 | 0 |
| 3.4-4.2 | 330 | 403 | 105 | 23 | 10 | 4 | 1 | 0 | 0 | 0 | 0 | 0 | 0 |
| 3.0-3.4 | 299 | 263 | 46 | 20 | 7 | 2 | 1 | 0 | 0 | 0 | 0 | 0 | 0 |
| 2.6-3.0 | 446 | 278 | 52 | 24 | 7 | 2 | 1 | 0 | 0 | 0 | 0 | 0 | 0 |
| 2.2-2.6 | 635 | 265 | 76 | 30 | 10 | 3 | 1 | 0 | 0 | 0 | 0 | 0 | 0 |
| 1.8-2.2 | 842 | 270 | 99 | 36 | 11 | 3 | 1 | 0 | 0 | 0 | 0 | 0 | 0 |
| 1.4-1.8 | 944 | 345 | 123 | 46 | 14 | 4 | 1 | 0 | 0 | 0 | 0 | 0 | 0 |
| 1.0-1.4 | 883 | 421 | 147 | 58 | 19 | 5 | 1 | 0 | 0 | 0 | 0 | 0 | 0 |
| 0.8-1.0 | 374 | 239 | 83 | 34 | 11 | 4 | 1 | 0 | 0 | 0 | 0 | 0 | 0 |
| 0.6-0.8 | 302 | 248 | 85 | 33 | 10 | 3 | 1 | 0 | 0 | 0 | 0 | 0 | 0 |
| 0.4-0.6 | 207 | 230 | 74 | 30 | 9 | 2 | 0 | 0 | 0 | 0 | 0 | 0 | 0 |
| 0.2-0.4 | 87 | 149 | 37 | 15 | 4 | 1 | 0 | 0 | 0 | 0 | 0 | 0 | 0 |
| 0-0.2 | 5 | 12 | 2 | 1 | 0 | 0 | 0 | 0 | 0 | 0 | 0 | 0 | 0 |

drop diameter (mm)

**Equipment 4: Prague Portable RS WSQ40**    total drop number    **10217**

| drop fall velocity (m s-1) | <0.5 | 0.5-1.0 | 1.0-1.5 | 1.5-2.0 | 2.0-2.5 | 2.5-3.0 | 3.0-3.5 | 3.5-4.0 | 4.0-4.5 | 4.5-5.0 | 5.0-5.5 | 5.5-6.0 | >6.0 |
|---|---|---|---|---|---|---|---|---|---|---|---|---|---|
| >10.0 | 0 | 0 | 0 | 0 | 0 | 0 | 0 | 0 | 0 | 0 | 0 | 0 | 0 |
| 9.0-10.0 | 1 | 0 | 0 | 0 | 0 | 0 | 0 | 0 | 0 | 0 | 0 | 0 | 0 |
| 8.2-9.0 | 1 | 0 | 0 | 0 | 0 | 0 | 0 | 0 | 0 | 0 | 0 | 0 | 0 |
| 7.4-8.2 | 2 | 0 | 0 | 0 | 0 | 0 | 0 | 0 | 0 | 0 | 0 | 0 | 0 |
| 6.6-7.4 | 3 | 0 | 0 | 0 | 0 | 0 | 0 | 0 | 0 | 0 | 0 | 0 | 0 |
| 5.8-6.6 | 6 | 0 | 0 | 0 | 1 | 1 | 1 | 1 | 0 | 0 | 0 | 0 | 0 |
| 5.0-5.8 | 17 | 1 | 2 | 9 | 4 | 3 | 3 | 1 | 1 | 0 | 0 | 0 | 0 |
| 4.2-5.0 | 60 | 5 | 36 | 19 | 9 | 6 | 3 | 2 | 1 | 1 | 0 | 0 | 0 |
| 3.4-4.2 | 243 | 124 | 125 | 35 | 19 | 7 | 4 | 2 | 1 | 1 | 0 | 0 | 1 |
| 3.0-3.4 | 287 | 222 | 75 | 26 | 10 | 4 | 2 | 1 | 1 | 1 | 0 | 0 | 0 |
| 2.6-3.0 | 447 | 397 | 61 | 24 | 10 | 4 | 3 | 1 | 1 | 1 | 0 | 0 | 0 |
| 2.2-2.6 | 708 | 601 | 64 | 23 | 10 | 5 | 2 | 1 | 0 | 0 | 0 | 0 | 0 |
| 1.8-2.2 | 1253 | 544 | 62 | 24 | 10 | 4 | 1 | 1 | 0 | 0 | 0 | 0 | 0 |
| 1.4-1.8 | 1490 | 377 | 58 | 19 | 7 | 3 | 1 | 0 | 0 | 0 | 0 | 0 | 0 |
| 1.0-1.4 | 1079 | 379 | 49 | 13 | 4 | 1 | 1 | 0 | 0 | 0 | 0 | 0 | 0 |
| 0.8-1.0 | 365 | 158 | 17 | 4 | 1 | 0 | 0 | 0 | 0 | 0 | 0 | 0 | 0 |
| 0.6-0.8 | 220 | 107 | 11 | 3 | 1 | 0 | 0 | 0 | 0 | 0 | 0 | 0 | 0 |
| 0.4-0.6 | 89 | 56 | 6 | 1 | 0 | 0 | 0 | 0 | 0 | 0 | 0 | 0 | 0 |
| 0.2-0.4 | 17 | 15 | 2 | 0 | 0 | 0 | 0 | 0 | 0 | 0 | 0 | 0 | 0 |
| 0-0.2 | 1 | 1 | 0 | 0 | 0 | 0 | 0 | 0 | 0 | 0 | 0 | 0 | 0 |

drop diameter (mm)




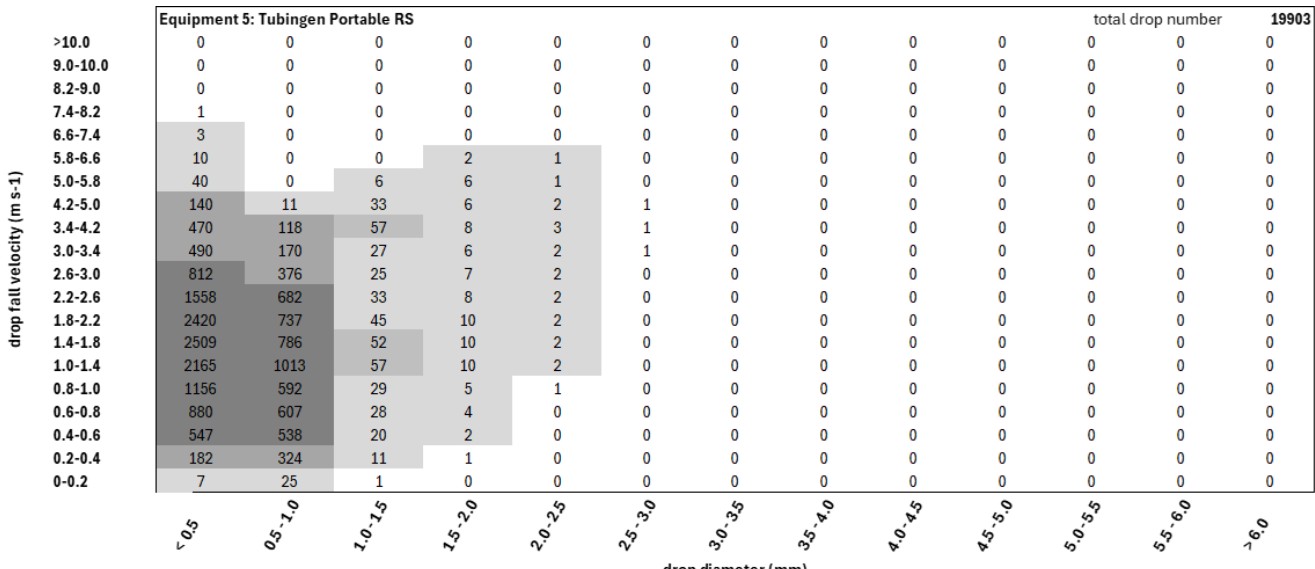

**Figures 8: Average drop size distribution and drop fall velocity for each rainfall simulator. The values shown correspond to mean values representing one-minute simulated rainfall. Each cell contains the total count of drops for each velocity-drop size class pair.**

### 3.2 Spatial rainfall distributions

The standard deviations of rainfall intensity across the five replicate measurements for each piece of equipment over the test area show distinct patterns. The maximum rainfall deviation is observed with Equipment 3 (Prague Portable Veejet 9550) at position E, where it is +80.7%, and the lower mean deviation is found in Equipment 5 (Tubingen Portable), which also has the lowest overall mean intensity (**Error! Reference source not found.**).









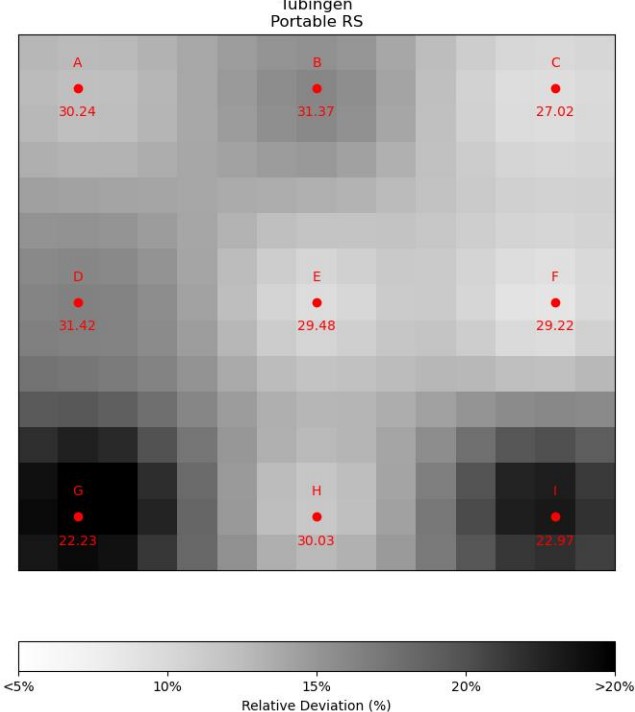

**Figure 9: Relative deviation of rainfall distribution (in %, measured values presented in mm h$^{-1}$; 5 replicates per measured position)**

## 3.3 Rainfall kinetic energy

The Prague Indoor Simulators show the highest KE-values, followed by the Prague portable simulators, and the Tübingen RS has the lowest values (Figure 10). Kinetic energy of rainfall shows significant differences for the simulators, with further results for the pairwise comparison given in the appendix (Table A3). Significant differences cannot be confirmed between the two nozzles used on the Prague Indoor and Portable Simulators. Except for the Tübingen Portable RS, the kinetic energy measured by the splash cups falls within the measured values by the LPM. We note that only one splash cup was monitored

for each position of each piece of equipment. In contrast, the LPM data were collected in five replicate one-minute intervals. At the same time, the splash cups summarized kinetic energy over periods of 10 to 20 minutes.



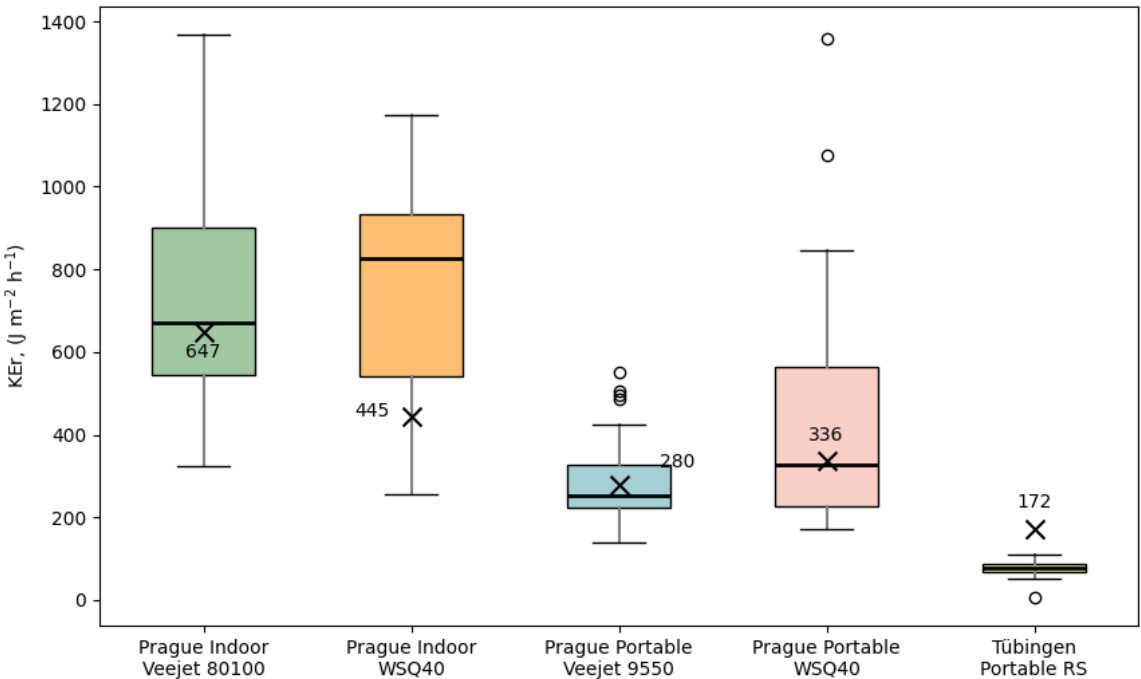

**Figure 10: Distribution of KEr of the five rainfall simulators (measured with LPM) used in the Kruskal-Wallis test (n = 225: KE$_r$, J m$^{-2}$ h$^{-1}$) KEr values calculated by the splash cups are shown as an "X" symbol**

The result for kinetic energy measured by both methods across the nine measured positions can be found in Figure A2. The values show that at low kinetic energy rainfall (defined as the lowest 25% of kinetic energy values in the dataset), the Tübingen splash cups consistently overestimated kinetic energy compared to the Thies disdrometers.

## 4      Discussion

For all simulations, more than 90% of measured drops were slower than 3.8 m s$^{-1}$, consistent with Iserloh et al. (2013), who
analyzed various European rainfall simulators. Drops exceeding velocities of 5.4 m s$^{-1}$ represent less than 1% of total drops for all simulations. The measurements' median drop diameters (d50) was smaller than 0.5 mm for all simulator-nozzle combinations. This is similar to those described by Petrů & Kalibová (2018) and smaller than in Ries et al. (2009); Fister et al. (2012); Iserloh et al. (2013). The formation of smaller drops has been attributed to the construction of the nozzle (Petrů & Kalibová, 2018). Median drop diameters in simulated rainfall are generally smaller than those observed in natural rainfall
(Assouline et al., 1997; Law et al., 2021).



The maximum rainfall deviation is observed with the Prague Portable Veejet 9550 at position E, where it is +80.7%. Although this is a considerably high value, it is lower than the maximum deviation found in other setups (Petrů & Kalibová, 2018). The Prague Portable WSQ40 has the highest overall deviation from the mean (33.2%). The deviation might be explained by the nozzles' physical characteristics and, potentially, by fluctuations in water pressure.

The devices' CU values range from 60.5 to 75.8%, and suggested CU values vary significantly across the literature. Little et al. (1993) and Keller & Bliesner (1990) suggest higher uniformity thresholds for values to be considered "good" or better (e.g., 80% and 84%, respectively) when addressing pressurized irrigation systems. Our CU values fall within the range of those presented by (Iserloh et al., 2013). The classification of spatial distribution commonly used in rainfall simulation studies was initially designed for pressurized irrigation systems (Keller & Bliesner, 1990; Little et al., 1993). Although the use of the
Christiansen Uniformity value is widely used, Green & Pattison (2022) have presented evidence of its limitations, including sensitivity to data resolution and varying layouts. We agree with Green & Pattison (2022) and Kubát et al. (2025) about its potential unsuitability for rainfall simulation studies, especially when presented by a single value unaccompanied by qualitative information on the spatial distribution patterns.

The drop size distribution found was similar to those presented by other authors (Iserloh et al., 2013; Petrů & Kalibová, 2018),
generally smaller than that observed in natural rain (Johannsen et al., 2020). However, for a higher falling distance of 5.5 meters applied, studies comparing simulated rainfall to natural rainfall have found similar terminal velocities (Assouline et al., 1997). Furthermore, we highlight that the size distribution found in our simulations (Figure A1) does not follow a monotonically decreasing function as described by Villermaux & Bossa (2009) but instead shows an increase until the 0.375–0.75 classes, followed by a decrease. However, we highlight the importance of defining class sizes in the size distribution curves, as a coarser class distribution may obscure finer details, particularly in the smaller drop-size classes, by aggregating a
range of drop sizes into broader categories.

We found a close relationship between the loss of sand and the kinetic energy of rainfall as measured by the disdrometer in four of the five equipment, in agreement with Scholten et al. (2011). However, the variability among sampling points, as measured by the T-cups, highlights the need for increased replication when using these devices. At the same time, this method
offers a cost-effective alternative to LPMs, making replication feasible. For low-kinetic-energy rainfall, the calibration of the T-cups requires further investigation: the 25% lowest kinetic energy registered by Thies was overestimated by the T-cups.

Determining rainfall kinetic energy is vital to understanding water erosion. Splash erosion, in particular, is the early stage of rain erosion, when, due to the kinetic energy of the rain, soil aggregates are fragmented and become more susceptible to displacement (Laburda et al., 2021), leading to soil loss when runoff begins, given there is a slope (Smith & Wischmeier,
1957). Higher kinetic energy has been associated with greater particle detachment (Fernández-Raga et al., 2010; Scholten et al., 2011). Rainfall kinetic energy is also a driving force behind interrill erosion (Zhang, 2019), and it is also considered in soil erosion models for water-driven processes, as it composes the calculation of rainfall-erosivity, particularly in models based on the USLE/RUSLE, such as the Soil and Water Assessment Tool (SWAT) (Arnold et al., 1998) and the Water and Tillage





Erosion and Sediment Model (Watem/Sedem) (Van Rompaey et al., 2001). Channel erosion, on the other hand, is primarily
driven by the intensity of rainfall and its resulting surface runoff, rather than kinetic energy (Vanmaercke et al., 2021).
Rainfall parameters have also been analysed across different surfaces, with vegetated surfaces being associated with larger
raindrop sizes and higher kinetic energy as compared to bare soil (Mosley, 1982; Vis, 1986). We highlight, however, that this
does not imply higher erosion rates, as the protective layer of plant litter acts as a buffer, absorbing the impact of raindrops
before they reach the soil surface (Seitz et al., 2015; Seitz et al., 2017; Senn et al., 2020). Our experiment underscores that
some of the most prominent advantages of portable rainfall simulators– their portability- also represent a significant limitation
– their small-scale operation, as already highlighted by others (Clarke & Walsh, 2007). Additionally, there is a noticeable
concentration of research in certain regions (Iserloh et al., 2013; Luz et al., 2024), which highlights the need to expand rainfall
simulation studies to areas critical for global security and vulnerable to erosion. Portable systems and splash cups hold the
potential to address these challenges by enabling cost-effective, flexible experimentation in underrepresented regions.
In this context, the portable simulators are designed for ease of transport and efficient field deployment. The Prague Portable
RS can be transported by a large vehicle, making it suitable for studies conducted in machinery-accessible fields. The simulator
requires a team of three to four people for optimal operation and can be fully assembled within one hour. The Tübingen
Portable RS is highly compact and lightweight and can be transported in two 20 kg standard suitcases. This makes it particularly
suitable for transport via commercial flights, allowing researchers to easily deploy it in remote or hard-to-reach locations. The
simulator can be quickly assembled by a team of two, making it a more convenient option for studies that demand mobility
and flexibility.

## 5    Conclusion

This work compared results from three rainfall simulators across five setups commonly used and compared the kinetic energy
of their rainfall spectra measured by a laser disdrometer and splash cups. The simulators delivered varying results, which is
attributed to their different structure in terms of transportability, among other things. We found that the portable systems used
have distinct lower kinetic energy characteristics compared to the indoor systems, but notable differences prevent direct
measurement comparisons.
The drop size distribution found aligns with those reported in previous and is smaller than those typically seen in natural rain.
However, the comparability and reproducibility within each individual device were given, which thus fulfills the main purpose
of rain simulation systems: the experimental reproducibility of rain conditions. Therefore, the rainfall simulators presented are
of great use for experimental field studies, and their application is recommended for imitation.
We confirm the pertinence of the splash cup method to assess rainfall kinetic energy as a reliable and cost-effective tool for
kinetic energy assessment, though calibration adjustments are needed for low-energy events.
As technology progresses, the boundary between aspiration and feasibility constantly evolves, opening new possibilities for
research and applied science. At present, the lack of comparability between measurements obtained from different rainfall



simulators remains a challenge for future research. The lack of comparability of the measurements from different rainfall simulators must be considered, and a higher degree of standardization would be desirable, as well as the ability to produce drops that better resemble those found in natural rain. Finally, we underscore the need to broaden the scope of studies into areas that are critical to the global food market and vulnerable to erosion, where tools like portable rainfall simulators and

splash cups could be valuable.



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





**Appendix A**

**Table A1: Descriptive statistics**

|  | Setup | Mean | St. Deviation | Median |
|---|---|---|---|---|
| **n of drops** | 1 | 14 682 | 3 165.7 | 13 837.0 |
| | 2 | 9 623 | 2 139.0 | 9 033.0 |
| | 3 | 10 678 | 2 331.5 | 10 595.0 |
| | 4 | 10 263 | 2 855.4 | 10 212.0 |
| | 5 | 19 902 | 5 161.4 | 21 118.0 |
| **Intensity (disdrometer)** | 1 | 65.2 | 20.2 | 58.4 |
| | 2 | 95.1 | 28.2 | 102.4 |
| | 3 | 64.5 | 22.1 | 60.0 |
| | 4 | 65.0 | 27.2 | 61.9 |
| | 5 | 28.2 | 4.8 | 29.4 |
| **KEr (Jm-2 h-1)** | 1 | 720.5 | 258.7 | 670.4 |
| | 2 | 753.3 | 262.4 | 826.6 |
| | 3 | 284.9 | 91.8 | 252.9 |
| | 4 | 415.8 | 260.3 | 326.4 |
| | 5 | 77.2 | 17.1 | 76.7 |
| **KE (Jm-2 mm-1)** | 1 | 11.0 | 2.1 | 10.8 |
| | 2 | 7.8 | 0.7 | 7.7 |
| | 3 | 4.5 | 0.6 | 4.5 |
| | 4 | 6.1 | 1.4 | 5.6 |
| | 5 | 2.8 | 0.7 | 2.7 |

Slight changes in the number of drops due to rounding may be observed when compared to the matrix of Velocities vs. Diameter (Figures
8).




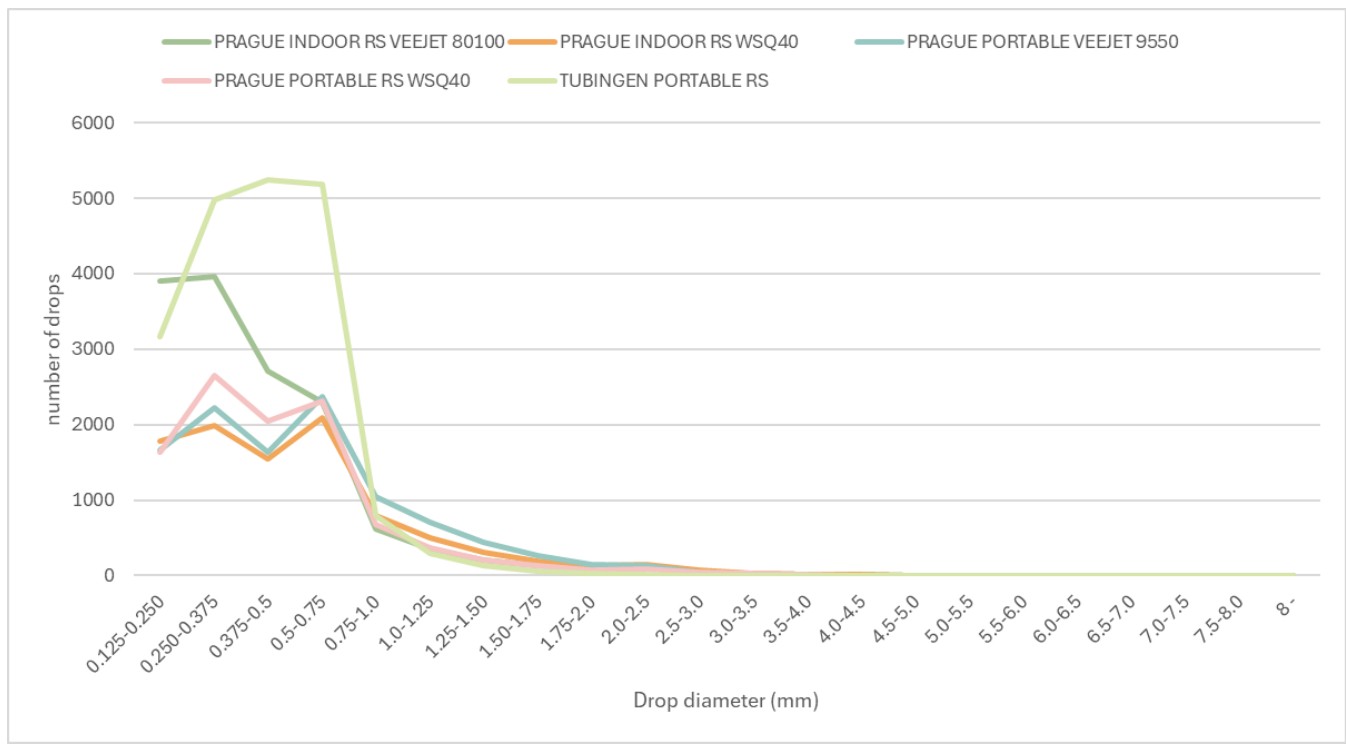

**Figure A1: Distribution of drop sizes for all five equipment**





**Table A2: Comparison between generated drops per equipment by pairwise method**

| Pair of equip. | Test Statistic | Standard Deviation of Test Statistic | Standardized Statistic | Sig. (p-value) | Adj. Sig.ᵃ (Bonferroni) |
|---|---|---|---|---|---|
| 1-2 | 81.48 | 13.72 | 5.94 | < 0.001 | 0.00 |
| 1-3 | 60.68 | 13.72 | 4.42 | < 0.001 | 0.00 |
| 1-4 | 69.26 | 13.72 | 5.07 | < 0.001 | 0.00 |
| 1-5 | -40.30 | 13.72 | 2.94 | 0.003 | 0.03 |
| 2-3 | -20.80 | 13.72 | 1.52 | 0.130 | 1.00 |
| 2-4 | -12.22 | 13.72 | 0.891 | 0.373 | 1.00 |
| 2-5 | -121.78 | 13.72 | inf | < 0.001 | 0.00 |
| 3-4 | 8.58 | 13.72 | 0.625 | 0.532 | 1.00 |
| 3-5 | -100.98 | 13.72 | 7.36 | < 0.001 | 0.00 |
| 4-5 | -109.56 | 13.72 | 7.99 | <0.001 | 0.00 |

Each row tests the null hypothesis that the distributions of each pair are equal. The asymptotic significance (two-tailed test) is displayed. The significance level is 0.05. The Bonferroni correction has adjusted the significance values for multiple tests.

**Table A3: Comparison between KEr per equipment by pairwise method**

| Sample 1 - Sample 2 | Test Statistic | Standard Deviation of Test Statistic | Standardized Statistic | Sig. (p-value) | Adj. Sig.ᵃ (Bonferroni) |
|---|---|---|---|---|---|
| 1-2 | -3.84 | 13.72 | -0.28 | 0.78 | 1.00 |
| 1-3 | 80.84 | 13.72 | 5.89 | < 0.001 | 0.00 |
| 1-4 | 57.71 | 13.72 | 4.21 | < 0.001 | 0.00 |
| 1-5 | 146.18 | 13.72 | 10.65 | < 0.001 | 0.00 |
| 2-3 | 84.69 | 13.72 | 6.17 | < 0.001 | 0.00 |
| 2-4 | 61.56 | 13.72 | 4.49 | < 0.001 | 0.00 |
| 2-5 | 150.02 | 13.72 | 10.93 | < 0.001 | 0.00 |
| 3-4 | -23.13 | 13.72 | -1.69 | 0.09 | 0.92 |
| 3-5 | 65.33 | 13.72 | 4.76 | < 0.001 | 0.00 |
| 5-4 | 88.47 | 13.72 | 6.45 | < 0.001 | 0.00 |

Each row tests the null hypothesis that the distribution of Sample 1 and Sample 2 are equal. The asymptotic significance (two-tailed test) is displayed. The significance level is 0.05.

a.  The significance values have been adjusted by the Bonferroni correction for multiple tests.





**Figure A2: Spatial distribution of KEr values for the five rainfall simulators across the nine measured positions (n = 225: KEr, J m⁻² h⁻¹) KEr values calculated by the splash cups are shown as an "X" symbol**