# Peer review of "Comparative Analysis of Compact Portable and Indoor Rainfall Simulators"

_EGUsphere, 2025_

## Author Comment (AC1)

**Response to Anonymous Referee #1 (RC1) to**

**preprint egusphere-2025-5908: "Comparative Analysis of Compact Portable and Indoor Rainfall Simulators"**

We thank you for the thorough review of our manuscript, the constructive comments, and the annotated PDF, which was very helpful in improving the manuscript.

| Reviewer comments | Authors responses |
|---|---|
| *"What specific problems does this study address? Each simulator operates under different conditions, which leads to discrepancies in the results. How can you justify or account for these discrepancies? No reference is made to natural rainfall conditions or the method used for measuring raindrop characteristics."* | This study is primarily intended to compare various portable systems and to assist scientists who wish to use small-scale rainfall simulators. There are no specific questions regarding the site used or erosion rates. It is a technical description of the possibilities for erosion research. Reference to natural rainfall conditions does thus not seem necessary. Further references on methodology can be added. |
| *"The abstract must clearly present the research significance, materials and methods, results, conclusions, and recommendations."* | We will streamline the structure and present clearer significance and research questions. |
| *"The text mentions five types of simulators... The type of simulator must be specified.*

*On what basis were these selected?"* | The rainfall simulators are presented in Table 1. There are three simulators, two of which were evaluated with two nozzle types, meaning a total of five devices. Each simulator has its own section (2.1.1 – 2.1.3) in the part "Equipment Design and Installation". However, we will provide additional information on the five types of rainfall simulators in the revised abstract.

These are the equipment currently employed at our institutions (CTU Prague, Czech 95 Republic; University of Tübingen, Germany). |
| *"Most of the information on the simulator designs is incomplete. The type of simulator, nozzle, and other details must be specified, and the results should be accurately presented for each simulator."* | Detailed descriptions of the rainfall simulators (including type, nozzle, and experimental setup) are provided in the "Equipment Design and Installation" section of the manuscript. Rainfall simulator-specific results can be found in the "Results" section.

We agree, however, that the abstract can better reflect this information. Accordingly, we will revise the abstract to include more information on the rainfall simulator types and to improve the presentation of the |

| | |
|---|---|
| | main results, while keeping the level of detail appropriate for an abstract. |
| *"The main reason for this study is not specified in the introduction."* | The study's primary objective is to identify similarities and differences among three rainfall simulators (five experimental setups) that are currently in use at our research institutions with respect to key rainfall characteristics. Additionally, the study introduces the use of Tübingen Splash Cups (T-cups) to estimate raindrop kinetic energy for the comparison of small rainfall simulators, which, to our knowledge, has not been previously applied in this context.

We will adjust the text in the "Introduction"section accordingly. |
| *"Some paragraphs are unnecessarily long in structure."* | We will review the text and consider this suggestion. |
| *"Failure to present the main hypothesis."* | We will consider this suggestion and adjust the text accordingly. |
| *"Repetition of objectives!"* | We will review the text accordingly. |
| *"The grammar and language of the manuscript should be improved."* | The manuscript was reviewed by a native English speaker and later processed with Grammarly. |
| *"New references should be used."* | Thank you for the recommendations. We will review the suggested references and include those we deem relevant to the present work |
| *"The introduction lacks a direct and quantitative link established between these technical differences and the final erosion outcomes (such as sediment yield or runoff). This creates a gap between "the characteristics of the simulated rainfall" and "its ultimate purpose (studying erosion)"."* | The aim of the study is to compare rainfall characteristics across five small portable rainfall simulator setups, rather than to assess erosion outcomes such as surface runoff or sediment yield. Consequently, a direct quantitative link between rainfall characteristics and erosion response is beyond the scope of this manuscript.

To address your concern, we will revise the introduction to more clearly explain this scope and to clarify that the characterization of simulated rainfall is intended as a methodological basis for interpreting erosion experiments conducted with these devices. |
| *"To what extent are the findings from these devices generalizable to the broader global community of rainfall simulators?"* | This is a good point, and we will add further outlook to the discussion and conclusion part. We believe the methodological description in our manuscript can serve as a standardized comparison procedure for small rainfall simulators. |
| *"Can general principles be derived from the results?"* | See above |
| *"Although "being the first" is an innovative point, the introduction* | We agree that novelty alone is insufficient and that the work's importance needs to be explained more clearly. In the revised introduction, we will explain that splash |

| | |
|---|---|
| *does not explain why this work is important.”* | cups are a simple, robust, and low-cost device for estimating raindrop kinetic energy. Their ease of use and affordability make them suitable for broader application and potential inclusion in standardized procedures for comparing rainfall characteristics across small portable rainfall simulators. |
| *“Why was this cup used and not other conventional methods?”* | From our perspective, splash cups are a conventional, standardized method for evaluating KE. Additionally, disdrometer measurements were used. |
| *“Why is comparing these three specific devices important?”* | These devices are independently developed, small, portable rainfall simulators currently in active use for erosion research. Results obtained with such devices are often compared across studies, although the rainfall characteristics produced by different simulators are rarely quantified using a common framework. By comparing these devices using a standardized procedure, the study highlights similarities and differences that are directly relevant for reproducibility, comparability, and interpretation of results obtained with small-scale rainfall simulators. We will clarify this motivation more explicitly in the revised introduction. |
| *“Equipment design and installation are missed.”* | The devices are presented in Table 1. There are three simulators, two of which were evaluated with two nozzle types, meaning a total of five devices. Each simulator has its own section (2.1.1–2.1.3) in Chapter 2, “Equipment Design and Installation”. Could the reviewer be more specific about what he considers in need of further description? |
| *“To what extent can environmental factors such as air humidity, water pressure, etc., be influential?”* | It is true that comparing measurements from different locations (and different times of year) will affect the outcome. However, this appears to be negligible for the comparison presented here. |
| *“It should be noted that these experiments were conducted in different regions, each with its own climatic parameters, which could affect the results.”* | See above |
| *“When the height is variable, the results will certainly change as well.”* | Absolutely. We addressed this in the discussion, and the results confirm that the indoor simulator achieves higher kinetic energy than the portable units. |
| *“In conditions outside of a rainfall simulator, how do you account for the effect of wind on raindrop characteristics?”* | As we stated in 2.1.2: “Wind can adversely affect the spatial distribution of rainfall, so we shielded the experimental plot with a plastic tarpaulin.”

In 2.1.3, we assumed the picture of the rainfall simulator would make it clear that the device is |

| | shielded from the wind; we will explicitly state this in the text. |
|---|---|
| *"The measurement of raindrop characteristics was not conducted under uniform conditions." 7:* | Since no erosion or runoff measurements were conducted or compared between devices, uniform rainfall conditions were not required for the aims of this study. |
| *"6: Has the plot effect been considered in this study?"* | With respect to plot effects, this study did not consider them because the focus was exclusively on rainfall characteristics, not on soil erosion responses. |
| *"Calculate other properties of raindrops, including area, perimeter, angle, and external energy."* | Regarding additional raindrop properties (e.g., area, perimeter, angle, or external energy), only parameters that could be robustly derived with the available measurement techniques were included. |
| *"8: This section remains unclear."* | We will revise the section "Statistical approach and data evaluation" to provide more detail on the statistical methods used and to improve transparency and readability. |
| *"Furthermore, there is no information on natural rainfall conditions."* | We will consider and include it. |
| *"The results are not very concise and clearly stated. It is necessary to provide detailed results of the characteristics of raindrops in each of the simulators."* | The main characteristics are presented in Table 2, Figures 7 – 10, and more detailed information is provided in the Appendix. Could the reviewer be more specific as to which information he considers should be included? |
| *"How do you compare the results with different rainfall intensities? The intensity of normal rainfall is still unknown."* | We do not compare rainfall intensities to natural rainfall, as the primary objective is comparability across devices to ensure replicable experimental conditions. The intensity of natural rainfall is therefore way to variable and changing in seconds during precipitation events. |
| *"The value of this coefficient is low. How do you justify it?"* | The aim of small-scale rainfall simulators is not to maximize spatial uniformity, but to provide repeatable rainfall conditions for relative comparisons under controlled settings. Similar CU values, and also a wide range of CU values, have been reported for comparable small scale rainfall simulators (Iserloh et al., 2013a).

In addition, CU is highly sensitive to the spatial resolution and layout of the sampling methodology, with the number and arrangement of collectors strongly influencing the resulting values under comparable rainfall conditions. CU should therefore be interpreted |

| | |
|---|---|
| | with caution and not overemphasized as a performance criterion (Green and Pattison, 2022) |
| *"In discussion section, what specific aspects of nozzle design might lead to smaller droplet production?"* | Smaller droplet production is primarily influenced by nozzle characteristics, including nozzle type, size, spray geometry, and operating pressure (Serio et al., 2025). We will clarify these aspects in the Discussion to explain how nozzle design can affect rainfall characteristics in small portable simulators. |
| *"How can these findings be used to improve the accuracy of future precipitation simulations?"* | This is a very good point, and we will further discuss improvements in the discussion part. |
| *"How do these deviations compare with similar results from other studies? Could the specific circumstances of this study be the main reason for these differences?"* | The aim was to keep circumstances mostly constant. However, simulator devices running in closed facilities and environments are not expected to exhibit extremely high variability between runs, at least not enough to significantly affect results. |
| *"Given the CU values, how can a more accurate or improved criterion be reached for evaluating the precipitation distribution in simulations?"* | A team, including some of the authors of this manuscript, has suggested using semi-variograms as a superior spatial and temporal metric compared to CU. We included this work in our discussion: Kubát, J.-F., Neumann, M., & Kavka, P. (2025). Semi-variograms provide superior spatial and temporal insights into artificial rainfall compared to Christiansen uniformity. *Journal of Hydrology*, 132740. https://doi.org/10.1016/j.jhydrol.2025.132740 |
| *"Could such differences have major impacts on soil erosion simulations and hydrological models? How can these differences be accounted for in future modeling?"* | The smaller drop sizes produced by small-scale rainfall simulators, compared with natural rainfall, can influence erosion processes. Because interrill erosion tends to dominate at this scale, sediment detachment may be underestimated relative to rill erosion from natural rainfall. However, small-scale simulators are primarily used for relative comparisons of treatments under controlled conditions, rather than for absolute predictions of field-scale erosion. |
| *"Are there more accurate methods for measuring kinetic energy of rain that could improve these analyses? What factors cause T-cups measurements to be inaccurate in low kinetic energy rain?"* | Yes, we will address this suggestion by mentioning other methods for measuring KE – and highlighting that getting KE from this is an indirect method, as it is done by the calculation from drop size distribution.

We hypothesize the factors that cause the overestimation of KE from the T-cups at low KE. We believe this could be an interesting future investigation. |
| *"Explain the relationship between kinetic energy of rain and soil erosion."* | A higher kinetic energy of raindrops will increase splash erosion, which means that the raindrop hits the soil with a higher force, leading to the detachment of soil particles and starting soil erosion. So, the higher the kinetic energy of raindrops, the higher is the potential for soil erosion. We will make this relationship clearer in the manuscript. |

| | |
|---|---|
| *"Given the high costs of conducting replicate experiments, what suggestions do you have for improving the accuracy and efficiency of these methods in the context of cost-oriented research?"* | Since T-cups are low-cost and easy to produce, increasing the number of replicates is feasible without substantially increasing overall cost. This enables effective capture of variability while keeping experiments affordable and efficient. |
| *"How can these effects be more accurately incorporated into soil erosion models to achieve better predictions of soil erosion?"* | This is an interesting question. We think that incorporating these surface dependent rainfall effects into soil erosion models could improve predictions, however, this topic is beyond the scope of the present study. Our focus is on characterizing rainfall characteristics produced by small scale rainfall simulators under controlled conditions, rather than on modeling soil erosion outcomes. |
| *"What solutions can be adopted to increase the scalability of these simulators without reducing accuracy and measurement capabilities? Can new technologies be used to develop these systems?"* | Increasing the scalability of small portable rainfall simulators is challenging, as larger scale can especially affect portability. However, there are examples of larger portable rainfall simulators that retain mobility and cover larger plots (1m × 3m; 2m × 8m). These larger devices have the advantage of being able to capture rill erosion in addition to interrill processes (Kavka et al., 2018; Iserloh et al., 2013b). |
| *"Study limitations and suggestions and future research should be included in the conclusion."* | We agree that the discussion of the study's limitations and the suggestion for future research could be more explicit and better structured. Thank you for the suggestion. |

Green, D. and Pattison, I.: Christiansen uniformity revisited: Re-thinking uniformity assessment in rainfall simulator studies, CATENA, 217, 106424, https://doi.org/10.1016/j.catena.2022.106424, 2022.

Iserloh, T., Ries, J., Arnáez, J., Boix-Fayos, C., Butzen, V., Cerdà, A., Echeverría, M., Fernández-Gálvez, J., Fister, W., and Geißler, C.: European small portable rainfall simulators: A comparison of rainfall characteristics, Catena, 110, 100–112, https://doi.org/10.1016/j.catena.2013.05.013, 2013a.

Iserloh, T., Ries, J., Cerdà, A., Echeverría, M., Fister, W., Geißler, C., Kuhn, N., León, J., Peters, P., Schindewolf, M., Schmidt, J., Scholten, T., and Seeger, M.: Comparative measurements with seven rainfall simulators on uniform bare fallow land, Zeitschrift für Geomorphologie, Supplementary Issues, 57, 11-26, 10.1127/0372-8854/2012/S-00085, 2013b.

Kavka, P., Strouhal, L., Jáchymová, B., Krása, J., Báčová, M., Laburda, T., Dostál, T., Devátý, J., and Bauer, M.: Double size fulljet field rainfall simulator for complex interrill and rill erosion studies, Stavební obzor - Civil Engineering Journal, 27, https://doi.org/10.14311/CEJ.2018.02.0015, 2018.

Serio, M. A., Caruso, R., Carollo, F. G., Bagarello, V., Ferro, V., and Nicosia, A.: The Hydraulic Assessment of a New Portable Rainfall Simulator Using Different Nozzle Models, Water, 17, 1765, https://www.mdpi.com/2073-4441/17/12/1765, 2025.

---

## Author Comment (AC3)

**Response to Jesús Rodrigo-Comino (RC3) to**

**preprint egusphere-2025-5908: "Comparative Analysis of Compact Portable and Indoor Rainfall Simulators"**

As the third reviewer, and having read the previous reviews from my colleagues, I will not repeat the comments made by Prof. Dunkerley, with most of which I agree. I think these points need to be better justified in the manuscript in order for the paper to be published with solid scientific support. In addition, having worked with some of the authors in the past, and with some of these or similar equipment, I trust the effort made by the authors when carrying out each rainfall simulation, especially those using large-size simulators. However, I believe that in its current form the paper cannot be published in a high-impact specialized journal and requires major revisions.

| Reviewer comments | Authors responses |
|---|---|
| *"The title is too broad: what type of soils, rainfall intensity, and geomorphological conditions are being addressed?"* | Thank you, we will consider. We did not address geomorphological conditions because the scope of the work was limited to evaluating the rainfall simulators, not measuring erosion on the plot. |
| *"Regarding the abstract, the very first sentence is too strong. Rainfall simulators are useful, but not indispensable."* | Thank you, we will rephrase it. |
| *"I do believe there are standardized methods, but not globally for all simulators. The authors state that they aim to clarify aspects of field work with rainfall simulators, but they do not specify what exactly is still unclear."* | Thank you, we will consider. |
| *"Several statements about disdrometers and calibration procedures refer to well-known issues and do not seem novel."* | Do you suggest we be more concise about these procedures? |
| *"I miss references to studies where these simulators have been applied in real conditions, in order to better understand how they perform and what results they produce."* | Thank you, we will incorporate the following studies: Gall et al. (2025); Riveras-Muñoz et al. (2025); Gall et al. (2024b); Gall et al. (2024a); Seitz et al. (2019); Seitz et al. (2017); Gall et al. (2022) |
| *"I do not find it appropriate to include figures taken far from the simulator, as they do not show its components, the fieldwork, or schematic diagrams with its characteristics. I would even consider including videos as supplementary material to validate the work and address some of the issues rightly pointed out by other reviewers. For example, in Figure 3, neither the simulator nor the soil surface can be seen."* | Thank you for the input. This is very helpful insight. |
| *"Not describing the soil type and its initial conditions is, in my opinion, a major flaw, as it* | We did not describe the soil type because the scope of the work was limited to |

| | |
|---|---|
| *prevents a full understanding of the experimental setup."* | evaluating the rainfall simulators, not measuring erosion on the plot. |
| *"The statistical analysis could be expanded slightly, including information on the libraries used and even the code to reproduce the figures, since the paper is presented as a study that should be reproducible by other researchers"* | Thank you, these will be included. |
| *"I do not fully understand why the figures are not properly numbered, for example Figure 8 (a, b, c)."* | Thank you, we will revise them. |
| *"There are many citation typos, likely related to Zotero or Mendeley, including missing references."* | Yes, this has been noted by the other reviewers. Thank you. |
| *"Personally, I do not like the black-and-white figures, particularly for the heatmaps, and I wonder whether this could be improved."* | We initially prepared coloured figures for a heatmap of rainfall intensities, but after deciding to present it as deviation from the mean, we thought a coloured scale could be misleading. |
| *"The discussion relies heavily on old references, and I do not clearly see the development of what is stated in the abstract and objectives, namely guiding authors towards standardized work and protocols. The objectives may need to be reformulated."* | Thank you. We will consider this to better align with the proposed objectives. |
| *"I have included additional comments and annotations in the PDF, specifically in the sections where I believe changes are necessary."* | Thank you, we will note those. |

.Gall, C., Nebel, M., Scholten, T., and Seitz, S.: The effect of mosses on the relocation of SOC and total N due to soil erosion and percolation in a disturbed temperate forest, Frontiers in Forests and Global Change, 7, 1–11, https://doi.org/10.3389/ffgc.2024.1379513, 2024a.

Gall, C., Nebel, M., Quandt, D., Scholten, T., and Seitz, S.: Pioneer biocrust communities prevent soil erosion in temperate forests after disturbances, Biogeosciences, 19, 3225–3245, https://doi.org/10.5194/bg-19-3225-2022, 2022.

Gall, C., Nebel, M., Scholten, T., Thielen, S. M., and Seitz, S.: Water's path from moss to soil Vol. 2: how soil-moss combinations affect soil water fluxes and soil loss in a temperate forest, Biologia, 1101–1113, https://doi.org/10.1007/s11756-024-01666-w, 2024b.

Gall, C., Oldenburg, S., Nebel, M., Scholten, T., and Seitz, S.: Effects of moss restoration on surface runoff and initial soil erosion in a temperate vineyard, SOIL, 11, 199-212, https://doi.org/10.5194/soil-11-199-2025, 2025.

Riveras-Muñoz, N., Seitz, S., Gall, C., Rodríguez, V., Witzgall, K., Kühn, P., Mueller, C. W., Oses, R., Seguel, O., Wagner, D., and Scholten, T.: Biocrusts as Climate-Dependent Regulators of Erosion, Water and Nutrient Cycling, Available as Preprint at SSRN, https://doi.org/10.2139/ssrn.5188208, 2025.

Seitz, S., Goebes, P., Puerta, V. L., Pereira, E. I. P., Wittwer, R., Six, J., van der Heijden, M. G. A., and Scholten, T.: Conservation tillage and organic farming reduce soil erosion, Agronomy for Sustainable Development, 39, 1-10, https://doi.org/10.1007/s13593-018-0545-z, 2019.

Seitz, S., Nebel, M., Goebes, P., Käppeler, K., Schmidt, K., Shi, X., Song, Z., Webber, C. L., Weber, B., and Scholten, T.: Bryophyte-dominated biological soil crusts mitigate soil erosion in an early successional Chinese subtropical forest, Biogeosciences, 14, 5775-5788, https://doi.org/10.5194/bg-14-5775-2017, 2017.